# The Effect of Cognitive Training with Neurofeedback on Cognitive Function in Healthy Adults: A Systematic Review and Meta-Analysis

**DOI:** 10.3390/healthcare11060843

**Published:** 2023-03-13

**Authors:** Yutaka Matsuzaki, Rui Nouchi, Kohei Sakaki, Jérôme Dinet, Ryuta Kawashima

**Affiliations:** 1Division of Developmental Cognitive Neuroscience, Institute of Development, Aging, and Cancer (IDAC), Tohoku University, Sendai 980-8575, Japan; 2Department of Cognitive Health Science, Institute of Development, Aging, and Cancer (IDAC), Tohoku University, Sendai 980-8575, Japan; 3Smart Aging Research Center (S.A.R.C.), Tohoku University, Sendai 980-8575, Japan; 4Department of Functional Brain Imaging, Institute of Development, Aging, and Cancer (IDAC), Tohoku University, Sendai 980-8575, Japan; 5Laboratoire Lorrain de Psychologie et Neurosciences de la Dynamique des Comportements (2LPN), Université de Lorraine, F-54000 Nancy, France

**Keywords:** cognitive training, neurofeedback, NIRS, cognitive function, memory

## Abstract

Background: Cognitive training aims to improve cognitive function through cognitive tasks or training games. Neurofeedback is a technique to monitor brain signals with either visual or auditory feedback. Previous studies suggest that a combination of cognitive training and neurofeedback has a superior effect on cognitive functions compared with cognitive training alone. However, no systematic reviews and meta-analyses of the benefits of cognitive training with neurofeedback (CTNF) exist. The purpose of this study was to examine the beneficial effects of CTNF in healthy adults using a systematic review and multilevel meta-analysis. Methods: PubMed, Scopus, PsychoINFO, and MEDLINE were searched for research papers reporting the results of interventions using CTNF. Results: After an initial screening of 234 records, three studies using near-infrared spectroscopy (NIRS) and one study using electroencephalography were extracted from the database. We performed a multi-level meta-analysis with three NIRS studies including 166 participants (mean ages ranged from 21.43 to 65.96 years). A multi-level meta-analysis revealed that CTNF has a beneficial effect on the episodic, long-term, and working memory domains. Conclusions: Although three studies were included in the systematic review and meta-analysis, our results indicate that CTNF using NIRS would lead to improvements in memory functioning.

## 1. Introduction

Cognitive function is one of the important factors for healthy living. It is associated with functional daily living and wellbeing [1,2], social networks [3], and life satisfaction [4]. For example, working memory capacity has been shown to help maintain focus and attention and engage in cognitively demanding activities [5]. In addition to those cognitive aspects, it has also been suggested to help maintain good emotional states through switching information about negative/positive emotions in working memory [6]. However, most cognitive functions, including working memory, begin to decline gradually after peaking in the 20s to 30s [7]. Therefore, research has been conducted on a method such as cognitive training, exercise [8], or nutritional [9] intervention to maintain and improve cognitive function.

Cognitive training (CT) aims to improve cognitive functions through cognitive tasks or training games. For example, cognitive tasks, such as memory tasks with progressively longer memory spans, are performed habitually using paper and pencil or a computer [10]. There are several types of CT, such as brain training [11,12], working memory training [13], and processing speed training [14]. A recent meta-analysis of studies reported that CT has a small to medium effect on the improvement of cognitive functions [15]. Behind this result, several factors, including individual characteristics such as age, cognitive function before the CT, and performance during training, have been suggested to lead to individual differences in the effects of the CT on cognitive function [16]. For this reason, studies have begun to examine factors that boost the effectiveness of CT.

Previous studies using neuroimaging have reported that brain activities during CT or cognitive tests have an important role in cognitive improvements [17,18,19]. For example, people who showed higher brain activities at the dorsolateral prefrontal cortex (DLPFC) during CT at the baseline received more benefit in processing speed and working memory from CT compared with those with lower brain activities [18]. In healthy older adults who showed high neural activity in the superior and middle frontal gyrus during memory tasks, it has been reported that working memory training is highly effective for performance gains [19]. Therefore, it expects that people will receive a more beneficial effect on cognitive function when they maintain greater brain activity throughout an intervention period.

Neurofeedback is a technique that uses visual or auditory feedback to monitor brain signals from electroencephalography (EEG), near-infrared spectroscopy (NIRS), or functional magnetic resonance imaging (fMRI) [20]. To infer activity in the brain, EEG captures action potentials on the surface of the brain; NIRS uses near-infrared light that penetrates the body to capture the hemoglobin density in the blood in the region of interest; and MRI uses a strong magnetic field [21]. In neurofeedback, it is assumed that some activity causes plastic changes in the neural bases used by that activity, resulting in functional changes [22]. Neurofeedback not only enables people to recognize their brain activities but also allows them to change them on their own [23].

According to recent research, combining different types of training benefits cognitive functions more than a simple training program [12,24,25]. Therefore, several researchers tried to combine CT and neurofeedback training, resulting in CT with neurofeedback (CTNF). In the typical CTNF, participants were asked to perform CT while maintaining greater brain activities. Several studies have reported that CTNF has a positive effect on cognitive functions [26,27]. For example, previous studies using NIRS reported that the CTNF showed greater beneficial effects on working memory, long-term memory, attention, and executive functions compared with CT alone in healthy adults [27,28]. These studies suggest that CTNF has a more positive impact on cognitive function than single domain training (e.g., CT or neurofeedback alone). Brain imaging devices such as NIRS are becoming smaller [29] and more resistant to movement [30], which can reduce artifacts during feedback. These features make it useful to combine CT with neurofeedback to improve the effectiveness of CT on cognitive function, and this is expected to increase in use in the future. However, no systematic review or meta-analysis study has been conducted to investigate the effects of CTNF on a wide range of cognitive functions. Summarizing previous findings on which cognitive domains CTNF is effective in may support user choice as brain imaging devices become more prevalent and are used in daily situations to maintain cognitive function. Here, we performed a multi-level meta-analysis to reveal the benefit of CTNF on cognitive functions in healthy adults.

## 2. Materials and Methods

### 2.1. Protocol and Registration

Analysis methods and eligibility criteria were specified in advance and documented in a protocol registered on PROSPERO (International Prospective Register of Systematic Review; Record ID = CRD42022341453). This systematic review and meta-analysis were performed in accordance with the PRISMA checklist [31]. Please see the Appendix A.

### 2.2. Eligibility Criteria

Our research question was, “Does CTNF enhance cognitive function in a healthy adult when compared with a single domain intervention?” Literature eligibility criteria were established based on the PICO criteria [32,33], and literature was searched. Population: healthy adults. Intervention: studies in which both CT and neurofeedback were given to participants. Comparison: Cognitive training or neurofeedback-only interventions. Sham feedback and active control are also included. Outcome: studies assessed cognitive function in participants before and after an intervention period. Research articles reported in academic journals written in English were included; reports on books and conference proceedings were excluded.

### 2.3. Search Strategy

The database was searched in May 2022. We used the following electronic databases to search for previous studies: PubMed, Scopus, MEDLINE, and PsycINFO. We added (“neurofeedback” OR “biofeedback”) and (“NIRS” OR “MRI” OR “EEG”) as keywords for the feedback study. In addition, (“cognitive training” OR “brain training” OR “cognitive function” OR “cognitive intervention” OR “video game” OR “gaming” OR “working memory training” OR “attention training” OR “processing speed training”) were added as keywords for the cognitive intervention domain. Finally, previous systematic review and meta-analysis were removed by the terms like AND NOT “systematic review” and AND NOT “meta analysis”.

### 2.4. Study Outcome

Cognitive function was categorized into seven different domains based on the previous studies [8,9,34]. Specifically, they were general cognitive function, episodic memory/long-term memory, working memory, short-term memory, attention, processing speed, and executive function. Episodic/long-term memory is the ability to store, integrate, and retrieve information over minutes, hours, days, and years [35], including the logical memory tasks of WMS-R and the verbal and visual memory tasks of the CNS vital sign [36]. Working memory is the ability to direct concentration to perform relatively simple manipulations, combinations, and transformations of information [35], and tasks such as digit span backward in WAIS-Ⅲ, n-Back task [37], the Sternberg task [38], and the working memory task from the CNS vital sign [36] were included. Short-term memory is the short-term encoding and maintenance of information in the consciousness [35], and digit span forward in WAIS-Ⅲ was classified. Attention includes control to maintain alertness, orienting necessary information from sensory input [39], and sustained-attention tasks in CNS vital signs [36] and D-CAT [40] were classified. Processing speed is the ability to perform simple cognitive tasks quickly and fluently [35], and symbol-digit coding tasks in WAIS-Ⅲ and processing speed tasks in CNS vital signs [36] were classified. Executive function is the cognitive function for controlling and combining information for goal-oriented behavior [41], and it includes a Stroop task [42], a trail-making test [43], a brief executive function task [26], MCAB [44], and an executive function task for CNS vital signs [36].

### 2.5. Data Extraction

Two investigators (Y.M. and R.N.) evaluated the content of each article and abstracted the demographic information, including the year of publication, study location, number of participants included, mean age, number and percentage of female participants, type of cognitive intervention, type of neurofeedback, comparison group, intervention period and duration, and all cognitive assessment scales.

### 2.6. Risk of Bias Assessment

The quality of the methodology in each study was assessed by two independent raters (Y.M. and R.N.) according to the physiotherapy evidence database (PEDro) scale [45,46]. This 10-item (and an additional item) scale was designed for the evaluation of methodological quality, including: (1) eligibility criteria were specified; (2) random allocation; (3) concealed allocation; (4) baseline comparability; (5) blinded participants; (6) blinded therapists; (7) blinded assessors; (8) adequate follow-up; (9) intention-to-treat analysis; (10) between-group comparisons; (11) point estimates and variability. If (2) through (11) apply, one point will be added for each, with a maximum of 10 points.

### 2.7. Data Synthesis and Statistical Analysis

To evaluate the range of scores on the cognitive assessment between the CTNF and comparison intervention groups, the standard mean difference (SMD) and standard deviation, or 95% confidence interval (CI), were used. Outcomes were grouped by cognitive domain (e.g., episodic/long-term memory, executive function, working memory, short-term memory, attention, processing speed, and visuo-spatial function) based on the descriptions in the previous studies used in the meta-analysis and the cognitive domain classifications from factor analysis of the relationships among cognitive functions [35].

Changes in scores from pre-intervention were used to perform meta-analyses, as they allowed for the comparison of more trials. If only pre- and post- data were given for a study, the change from the baseline score was calculated by subtracting the score in pre-intervention from the score in post-intervention. The following equation was used to convert standard error (SE) into standard deviation (SD):SD = SE × √N

Changes from pre-intervention standard deviation (SD) were calculated following equation:SD_E/change_ = √SD^2^_E/baseline_ + SD^2^_E/final_ − (2 × 0.5 × SD_E/baselime_ × SD_E/final_)

All meta-analyses were conducted to summarize the effect sizes using the meta package [47] and metafor package [48] in R (https://www.R-project.org/, accessed on 25 July 2022). Statistical heterogeneity among the trials was assessed, and a *p* value of <0.05 was regarded as statistically significant. The positive impact on cognitive function after the intervention was expressed as a positive effect estimate. A negative estimate showed poorer cognitive task performance after the intervention. For indicators that worsen with higher values (e.g., error rate in the mental rotation test), positive and negative values were reversed and converted so that a positive value indicates an effect. A random-effects model was used because heterogeneity between trials was expected. Heterogeneity was assessed using Higgins I^2^ statics. If I^2^ was greater than 40%, the studies included in the analysis were considered to be heterogeneous. To present the results graphically, forest plots were used.

## 3. Results

### 3.1. Literature Search and Study Selection

From the literature search, a total of 387 abstracts were identified. After excluding duplicates (*n* = 153), 234 articles remained. The 234 studies were then screened by titles and abstracts, leaving 100 studies. Of the 100 studies, 95 were excluded for the following reasons: 41 studies focused only on patient participants; 19 studies used neurofeedback intervention only; nine studies used neurofeedback with non-cognitive intervention (e.g., meditation); one study used CT only; 20 studies were non-interventional studies; and five studies were not written in English. Five studies remained, one of which did not provide the necessary statistics for the meta-analysis. Finally, we found four studies using CTNF. The selection process above is depicted in the PRISMA flow chart (Figure 1).

### 3.2. Characteristics of Included Studies

There were a total of 306 participants in four eligible studies (Table 1). The previous studies included were conducted in Japan, Israel, and the USA. The sample size ranged from 20 to 140. The sample’s mean age ranged from 21.43 to 65.96, with females accounting for 39.29% to 75.58% of the total. Regarding the type of cognitive intervention, two studies used multi-domain (e.g., memory and processing speed) intervention [27,49], and the other two studies used working memory training [26,28]. Three studies used NIRS for neurofeedback [27,28,49], and only one study used EEG for neurofeedback [26]. For the control group, one study used CT with SHAM neurofeedback [28]. All three remaining studies employed an active control group, with another group receiving CT or neurofeedback alone [26,27,49]. The average intervention time per session ranged from 21.25 to 30 min (10 to 60 min), from four sets over two weeks to every day for four weeks. Since there was only one study that used EEG for CTNF, we used three NIRS studies in the meta-analysis (*n* = 166). The quality assessment score on the PEDro scale ranged from 3 to 9, with an average of 5.33 (SD = 3.21).

### 3.3. Episodic Memory/Long-Term Memory

Two studies provided results regarding episodic/long-term memory (Figure 2A). Of those two studies, Nouchi et al. (2022) [27] provided results for the episodic memory test of the WMS-R. Accevedo et al. (2022) [49] provided the visual/verbal memory test from the CNS vital sign [36]. Meta-analysis using all studies revealed that CTNF showed a statistically significant positive effect on episodic/long-term memory (I^2^ = 16.69%, SMD = 0.38, 95% CI 0.12 to 0.64).

### 3.4. Executive Function

Three studies provided the results about tasks related to executive function (Figure 2B). Two studies used the Stroop task [27,28], and one [49] used the executive function task in the CNS vital sign [36]. Hosseini et al. (2016) [28] also used the trail making test and MCAB. We did not find any statistically significant result from meta-analysis (I^2^ = 0%, SMD = 0.21, 95% CI −0.04 to 0.46).

### 3.5. Working Memory

All studies included in the meta-analysis provided results regarding working memory (Figure 2C). Of those three studies, (a) one study provided results for the digit span task [27], (b) one for the n-back task and Sternberg task [28], and (c) one study [49] used a working memory task in the CNS vital sign [36]. Meta-analysis showed a statistically significant improvement in working memory (I^2^ = 20.55%, SMD = 0.39, 95% CI 0.05 to 0.72).

### 3.6. Short-Term Memory

One study provided results about short-term memory (Figure 3A). Nouchi et al. (2022) [27] used the digit span forward task. We did not find any statistically significant result from the meta-analysis using all studies on short-term memory (I^2^ = 0%, SMD = −0.19, 95% CI −0.63 to 0.25).

### 3.7. Processing Speed

Two studies reported results regarding processing speed (Figure 3B). Nouchi et al. (2022) [27] provided the results of the digit symbol coding task, and Acebedo et al. (2022) [49] provided the processing speed task from the CNS vital sign [36]. We did not find any statistically significant result from the meta-analysis using all studies on processing speed (I^2^ = 55.09%, SMD = 0.13, 95% CI −0.40 to 0.67).

### 3.8. Attention

Two studies reported the results about attention (Figure 3C). One study [27] reported the results of the digit cancellation task, and the other study [28] provided the attention task from the CNS vital sign [36]. We did not find any statistically significant result from the meta-analysis using all studies on attention (I^2^ = 66.04%, SMD = 0.53, 95% CI −0.11 to 1.17).

### 3.9. Visuo-Spatial Performance 

Only one study reported results about visuo-spatial performance (Figure 3D). Nouchi et al. (2022) [27] used the mental rotation test. We did not find any statistically significant result from meta-analysis (I^2^ = 0%, SMD = −0.27, 95% CI −0.72 to 0.17).

### 3.10. Quality Assessment

Table 2 shows an assessment of the methodological quality of the previous studies used in the meta-analysis. For question 11 (point estimates and variability), all three studies did not report line estimates of the main outcome variable. Concealed assignment of groups (Q3) and whether the study was blinded to participants (Q5), therapists (Q6), and examiners (Q7) were scored low. On the contrary, all studies had an intention to treat analysis (Q9) and reported the results of statistical comparisons between the CT, neurofeedback, and control groups (Q10).

## 4. Discussion

In this meta-analysis and systematic review, we investigated whether CTNF has a beneficial effects on cognitive functions. This multi-level meta-analysis, which included three studies, first revealed that CTNF significantly improved working and episodic memory/long-term memory performances compared with other training groups.

The first main finding is that CTNF has beneficial effect on working memory performance. This result is in line with previous studies using working memory training. One of the included studies in our meta-analysis directly used working memory training as CT [28]. In addition, the other studies used working memory training as one of multiple CT programs [27,49]. Previous studies have reported that working memory training improved working memory performance compared with other types of CT or control groups [15,50]. The findings are also consistent with the previous studies showing that frontal lobe activity early in training measured by NIRS predicted improved working memory performance after cognitive training [18,19]. Therefore, CTNF would have a beneficial effect on working memory performance. However, this study first revealed that CTNF has a superior effect on improving working memory performance compared with CT alone or active control training groups using meta-analysis.

The second main finding is that CTNF improved episodic memory/long-term memory performance. This result is consistent with previous literature using neurofeedback [51] and CT alone [52]. A previous meta-analysis study revealed that neurofeedback training alone, using EEG, improved episodic memory performance [53]. In addition, other meta-analyses using CT alone have reported that CT improves episodic memory performance compared with no-intervention groups [52]. However, this study showed the first scientific evidence that CTNF using NIRS has a positive impact on episodic memory/long-term memory performance compared with CT alone and the active control group using meta-analysis. 

Our meta-analysis revealed that CTNF has positive effects on memory functioning (working memory and episodic memory/ long-term memory). It should be noted that CTNF has greater effects on cognitive improvements compared with CT alone and active control groups. It indicates that neurofeedback would have an important role in the improvements in memory functioning after CTNF. All of the included studies in our meta-analysis measured brain activities at DLPFC, and the results were given as feedback to participants. Previous neuroimaging studies have reported that the DLPFC plays an important role in working and episodic memory performance [54,55]. One previous study revealed that the greater DLPFC activations during CT were associated with greater cognitive benefits [18]. The DLPFC has been found to play a critical role in the use of strategies for memory encoding and retrieval [54]. These characteristics of the DLPFC are thought to work well for both working memory and long-term memory, and thus feedback of neural activity in the DLPFC during cognitive training may have led to improved memory performance. Based on the present findings, we make the following speculation: The CTNF group is asked to monitor the DLPFC activities. In addition, they try to increase brain activities at the DLPFC. The CTNF can maintain greater brain activity at the DLPFC during training compared with the CT alone or the active control groups. The DLPFC plays a critical role in the use of memory encoding and retrieval strategies. Therefore, the CTNF group has superior beneficial effects on memory functioning compared with the other groups. 

In two of the studies included in our meta-analysis, training duration was short (10 to 20 min) but relatively high frequency interventions: 5 or more days per week [27,49]. The one remaining study, on the other hand, was conducted four times in two weeks and led to improved cognitive function [28]. It is not clear from this meta-analysis whether the frequency of interventions is optimal in terms of cost to the subject and benefit to cognitive function. It is noted, however, that NIRS has the advantage of being non-invasive and easy to set up compared with other imaging methods, and it has the advantage of easily providing real-time brain activity feedback [29,30]. Ease of use and the advantage of not having to increase cognitive training time simply because it is based on real-time or immediate feedback may work well for combined training. Although our meta-analysis was conducted on healthy adults, this feature is beneficial in children and patient groups, who often have short concentration periods and physical movement.

This study has some limitations. First, we only included three studies using NIRS for the meta-analysis. For the reason that only three studies were included in the meta-analysis, the reliability and generality of the results are limited. Although heterogeneity was low in the cognitive domains where the CTNF effect was observed, but in general, the small number of data points may have biased the results. Since NIRS was the only instrument used for CTNF in our meta-analysis, it is expected that studies using other imaging instruments would be accumulated. Second, we did not perform a subgroup analysis by subject demographics and indicators of cognitive test. The subgroups in which CT is found to be effective are still debated [16]. Although it is important to examine factors such as the effects of aging and participant demographics on the effects of CTNF, a meta-regression analysis was not conducted due to the small number of previous studies in our meta-analysis. It is also possible that more detailed results will be obtained when the cognitive test is classified into a narrow domain [35] instead of the broad domain, which was used in this study. Third, our meta-analysis included only healthy adults. It is necessary to examine whether similar results can be obtained for patients or subclinical groups. For example, CTNF studies with children with attention deficit/hyper activity disorder have shown a reduction in their symptoms and problems related to executive function and working memory [56,57], an area in which CTNF is expected to be applied. Fourth, the results of the quality assessment using the PEDro scale indicated that the studies included in our meta-analysis were of low quality as RCTs, in particular, in terms of blinding the study. The quality of RCTs examining the effects of CTNF on cognitive function would need to be improved.

## 5. Conclusions

This study conducted a systematic review and meta-analysis of the effects of CTNF on a wide range of cognitive functions in healthy adults. The current meta-analysis revealed that CTNF with NIRS enhanced working memory and episodic memory/long-term memory performances compared with other intervention groups. In two of the three included studies, participants were asked to maintain greater activity in the DLPFC during the CTNF. It indicates that maintaining greater brain activities at the DLPFC would be an important factor for boosting cognitive function, especially in the memory domain, after the interventions. The current study has some limitations. However, this study was the first to demonstrate with scientific evidence that CTNF has a superior effect on cognitive improvements in healthy adults compared with other types of intervention training.

## Figures and Tables

**Figure 1 healthcare-11-00843-f001:**
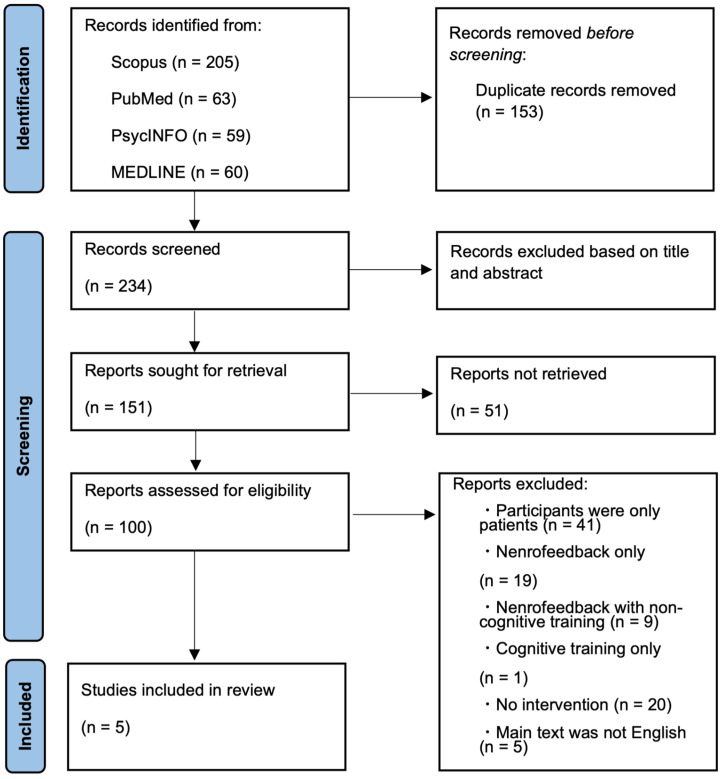
PRISMA flow chart.

**Figure 2 healthcare-11-00843-f002:**
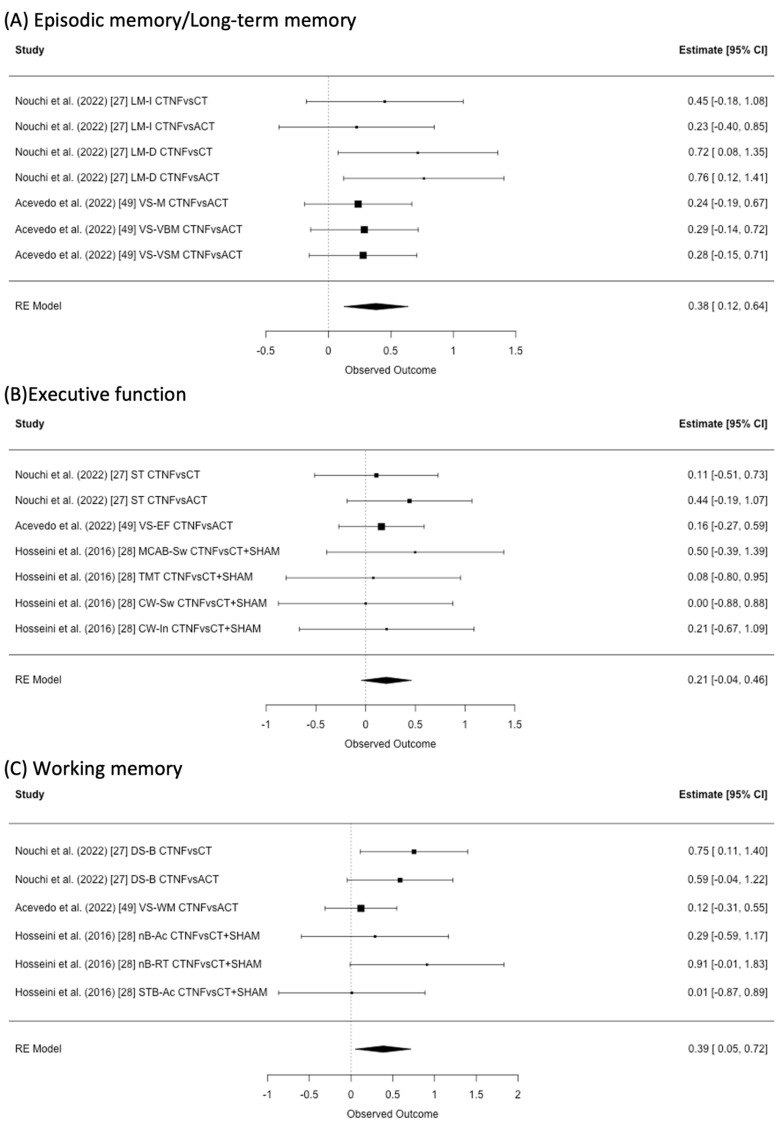
Effect estimates and a forest plot of cognitive training with neurofeedback (CTNF) on (**A**) episodic/long-term memory, (**B**) executive function, and (**C**) working memory. CT: cognitive training, ACT: active control, SHAM: sham feedback, LM-I: long term memory immediate recall, LM-D: long-term memory delay recall, VS-M: CNS vital sign memory, VS-VBM: CNS vital sign verbal memory, VS-VSM: CNS vital sign visual memory, ST: Stroop task, VS-EF: CNS vital sign executive function, MCAB-Sw: mobile cognitive assessment battery switching, TMT: trail making test, CW-Sw: color-word interference test switch, CW-In: color-word interference test inhibit, DS-B: digit span backward, VS-WM: CNS vital sign working memory, nB-Ac: N-back accuracy, nB-RT: N-back response time, STB-Ac: Sternberg task accuracy [27,28,49].

**Figure 3 healthcare-11-00843-f003:**
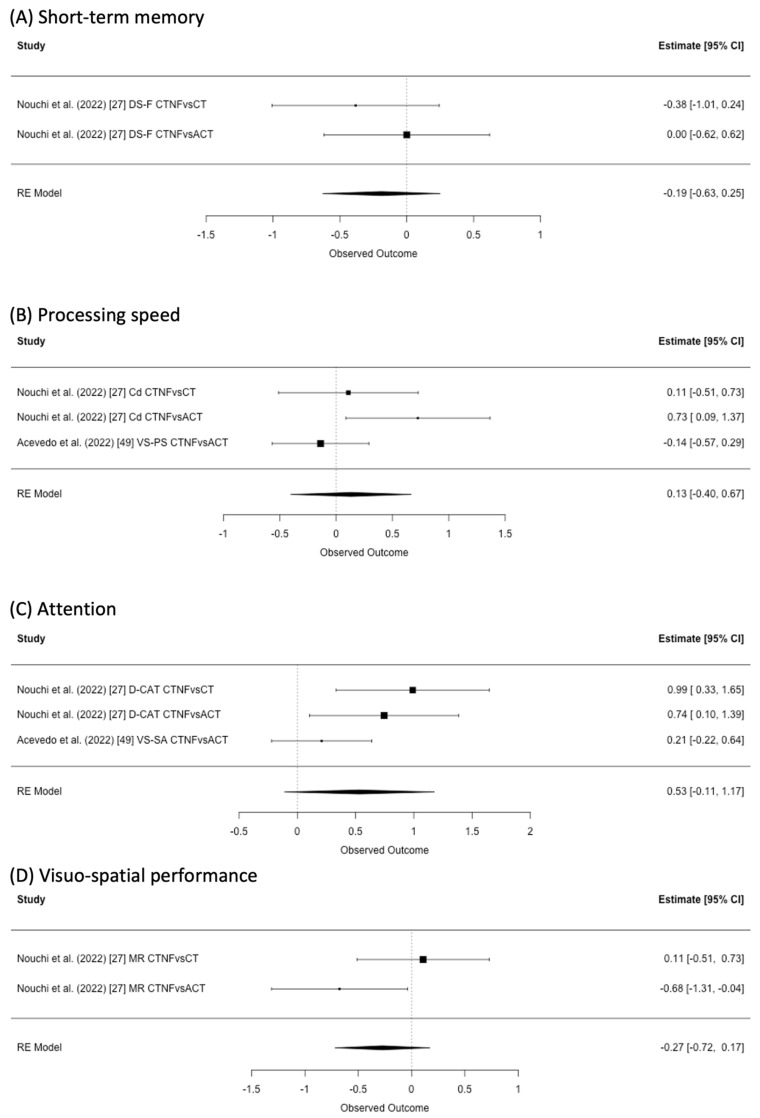
Effect estimates and forest plot of cognitive training with neurofeedback (CTNF) on (**A**) short-term memory, (**B**) processing speed, (**C**) attention, and (**D**) visuo-spatial performance. CT: cognitive training, ACT: active control, DS-F: digit span forward, Cd: digit symbol coding, VS-PS: CNS vital sign processing speed, D-CAT: digit cancellation task, VS-SA: CNS vital sign sustained attention, MR: mental rotation [27,49].

**Table 1 healthcare-11-00843-t001:** Study characteristics of included studies.

Study and Year	Country	No. of Participants	Mean Age	Female (%)	Domain of Cognitive Training	Type of Neurofeedback	Brain Regions Used for Neurofeedback	Comparison Groups	Intervention Period	Intervention Frequency (Time per Session)	Study Outcomes
Nouchi et al., 2022 [27]	Japan	60	21.43	50	Processing speed, Memory span, Attention	NIRS (2ch)	Left and right DLPFC	CT, ACT (Tetris)	Four weeks	Every day (20 min)	Cd, ST, D-CAT, DS-F, DS-B, LM-I, LM-D, MR
Acevedo et al., 2022 [49]	USA	86	65.96	75.58	Multiple domain	NIRS (1ch)	Right prefrontal cortex	ACT (Tetris)	Four weeks	5–7 days/week (10–15 min)	VS-VBM, VS-VSM, VS-M, VS-PS, VS-EF, VS-WM, VS-SA
Gordon et al., 2020 [26]	Israel	140	22.08	39.29	Working memory	EEG (3ch)	Parietal midline (Pz)	NFT + ACT (VS), NFT, WMT, ACT (VS), Silent Control	Five weeks	2 times/week (30–60 min)	Executive function and mental rotaion test
Hosseini et al., 2016 [28]	USA	20	24.6	50	Working memory	NIRS (52ch)	Left and right DLPFC	CT + SHAM feedback	Two weeks	4 sessions (25 min)	nB, TMT, CW, MCAB-Sw, STB

Note: NIRS: near-infrared spectroscopy, EEG: electroencephalography, DLPFC: dorsolateral prefrontal cortex, CT: cognitive training, ACT: active control, NFT: neurofeedback training, VS: visual search training, WMT: working memory training, Cd: digit symbol coding, ST: Stroop task, D-CAT: digit cancellation task, DS-F: digit span forward, DS-B: digit span backward, LM-I: long term memory immediate recall LM-D: long-term memory delay recall, MR: mental rotation, VS-VBM: CNS vital sign verbal memory, VS-VSM: CNS vital sign visual memory, VS-M: CNS vital sign memory, VS-PS: CNS vital sign processing speed, VS-EF: CNS vital sign executive function, VS-WM: CNS vital sign working memory, VS-SA: CNS vital sign sustained attention, nB: N-back task, TMT: trail making test, CW: color-word interference test, MCAB-Sw: mobile cognitive assessment battery switching, STB: Sternberg task.

**Table 2 healthcare-11-00843-t002:** Quality assessment scores of studies for meta-analysis using the PEDro scale.

No.	Items	Nouchi et al. (2022) [27]	Acevedo et al. (2022) [49]	Hosseini et al. (2016) [28]
1	Eligibility criteria were specified (additional)	Yes	No	Yes
2	Subjects were randomly allocated to groups (in a crossover study, subjects were randomly allocated an order in which treatments were received	Yes	Yes	No
3	Allocation was concealed	Yes	No	No
4	The groups were similar at baseline regarding the most important prognostic indicators	Yes	No	Yes
5	There was blinding of all subjects	Yes	No	No
6	There was blinding of all therapists who administered the therapy	Yes	No	No
7	There was blinding of all assessors who measured at least one key outcome	Yes	No	No
8	Measures of at least one key outcome were obtained from more than 85% of the subjects initially allocated to groups	Yes	No	Yes
9	All subjects for whom outcome measures were available received the treatment or control condition as allocated, or, where this was not the case, data for at least one outcome was analyzsed by “intention to treat”	Yes	Yes	Yes
10	The results of between-group statistical comparisons are reported for at least one key outcome	Yes	Yes	Yes
11	The study provides both point measures and measures of variability for at least one key outcome	No	No	No
	Total (Max 10)	9	3	4

## Data Availability

Not applicable.

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
