# Peer review of "The Effect of Cognitive Training with Neurofeedback on Cognitive Function in Healthy Adults: A Systematic Review and Meta-Analysis"

_healthcare, 2023, doi:10.3390/healthcare11060843_

Round 1

Reviewer 1 Report

Matsuzaki et al wrote a review on the effect of cognitive training on cognitive fx in healthy adults. CTNF was reported to have positive effects on memory functioning that is aligned with brain functioning findings. Overall, this is a commendable effort.

The methods was clearly stated, in terms of the search strategy, checked against PRISMA standards and registered on PROSPERO. Quality assessment was also conducted using PEDro scale. This is good practice

Feedback:

Study outcome was categorized into 7 different domains based on previous studies
-how were some of the various assessment harmonized when used as an outcome in the MA
-each of the task may appeared to be measuring different aspect of the same cognitive domain for instance, how do authors reconcile this?

Based on the limitations listed, could there be some improvements that can be made by the authors?
-broadening eligibility criteria of studies (so that it is not limited to 3 (?)) this is too few to make any generalizations or meaningful comparisons
-conducting similar analysis but in patient group (?)

The discussion section needs to be further elaborated and explained at greater length in terms of the research question which the review aims to address, below are some scope that can be further expanded in discussion
-based on the findings, so what? what can be improved? How can this be applied?
-based on the studies, what were the common characteristics behind the success or effective intervention-did the intervention period/frequency matter? 

-what were intervention (i.e CTNF) that yield more promising results?

-Were there differences attributed to patient/age/gender etc?

-is findings dependent on NIRS and EEG? or the brain imaging method used?

Author Response

Dear Reviewer

Thank you for the comments to our manuscript.

We revised introduction, methods, and discussion section following your comments.

The following are the revisions. We would appreciate it if you could check them.

The methods was clearly stated, in terms of the search strategy, checked against PRISMA standards and registered on PROSPERO. Quality assessment was also conducted using PEDro scale. This is good practice

Feedback:

Study outcome was categorized into 7 different domains based on previous studies
-how were some of the various assessment harmonized when used as an outcome in the MA

Response: Thank you for your comment. As used in previous studies 1,2, the indicators from each study are grouped by broad cognitive domain for meta-analysis. Cognitive domain classifications are based on factor analysis of the relationships among cognitive functions 3. As you pointed out, this point was not clear, so we have added a description in 2.7. (Lines 127-132)

“Outcomes were grouped by cognitive domain (e.g., episodic/ long-term memory, executive function, working memory, short-term memory, attention, processing speed, and visuo-spatial function) based on the description in the previous studies used in the meta-analysis and the cognitive domain classifications from factor analysis of the relationships among cognitive functions [38].”

Reference

(1)      Lampit, A.; Hallock, H.; Valenzuela, M. Computerized Cognitive Training in Cognitively Healthy Older Adults: A Systematic Review and Meta-Analysis of Effect Modifiers. PLoS Med. 2014, 11 (11). https://doi.org/10.1371/journal.pmed.1001756.

(2)      Bonnechère, B.; Langley, C.; Sahakian, B. J. The Use of Commercial Computerised Cognitive Games in Older Adults: A Meta-Analysis. Sci. Rep. 2020, 10 (1), 1–14. https://doi.org/10.1038/s41598-020-72281-3.

(3)      Schneider, J.; McGrew, K. The Cattell-Horn-Carroll (CHC) Model of Intelligence. In Contemporary Intellectual Assessment: Theories, Tests, and Issues; Flanagan, D., Harrison, P., Eds.; Guilford: NewYork, 2012.

-each of the task may appeared to be measuring different aspect of the same cognitive domain for instance, how do authors reconcile this?

Response:  We thank the reviewer for this comment. Previous study suggested that same cognitive domain would be divided into sub-domains. For example, the executive function would be divided into switching and inhibition are assumed. However, since a relatively strong relationship is assumed between these subcomponents 4, there is a meta-analysis that summarizes the data by cognitive domain as mentioned in the previous comment, and no problems were found in the heterogeneity indices in the analysis of this study (except for attention and processing speed, which did not differ significantly), we did not think a problem with the classification in this study.

Based on your comment, we have added a description of this point int the limitation. (Lines 319-221)

“It is also possible that more detailed results will be obtained when the cognitive test is classified into narrow domain [38] instead of broad domain which was used in this study. “

Reference

(4)      Miyake, A.; Friedman, N. P.; Emerson, M. J.; Witzki, A. H.; Howerter, A.; Wager, T. D. The Unity and Diversity of Executive Functions and Their Contributions to Complex “Frontal Lobe” Tasks: A Latent Variable Analysis. Cogn. Psychol. 2000, 41 (1), 49–100. https://doi.org/10.1006/cogp.1999.0734.

Based on the limitations listed, could there be some improvements that can be made by the authors?
-broadening eligibility criteria of studies (so that it is not limited to 3 (?)) this is too few to make any generalizations or meaningful comparisons

Response: Thank you for your comments. Our meta-analysis included the small number studies. It is one of the limitations of this study. However, there are examples of meta-analyses reported using as few as three or four studies5–7, so we do not think that this necessarily disqualifies the meta-analysis. Based on your suggestion, we have revised the description of the limitations and emphasized this point. (Lines 309-312).

“Because only three studies were included in the meta-analysis, the reliability and generality of the results is limited. Although heterogeneity was low in the cognitive domains where the CTNF effect was observed, but in general, the small number of data may have biased the results.”

Reference

(5)      Maitra, P.; Caughey, M.; Robinson, L.; Desai, P. C.; Jones, S.; Nouraie, M.; Gladwin, M. T.; Hinderliter, A.; Cai, J.; Ataga, K. I. Risk Factors for Mortality in Adult Patients with Sickle Cell Disease: A Meta-Analysis of Studies in North America and Europe. Haematologica 2017, 102 (4), 626–636. https://doi.org/10.3324/haematol.2016.153791.

(6)      Valkanova, V.; Ebmeier, K. P.; Allan, C. L. CRP, IL-6 and Depression: A Systematic Review and Meta-Analysis of Longitudinal Studies. J. Affect. Disord. 2013, 150 (3), 736–744. https://doi.org/10.1016/j.jad.2013.06.004.

(7)      Goldberg, S. B.; Pace, B. T.; Nicholas, C. R.; Raison, C. L.; Hutson, P. R. The Experimental Effects of Psilocybin on Symptoms of Anxiety and Depression: A Meta-Analysis. Psychiatry Res. 2020, 284 (April 2019), 112749. https://doi.org/10.1016/j.psychres.2020.112749.

-conducting similar analysis but in patient group (?)

Response: We wish to thank the reviewer for this comment. The purpose of this study was a meta-analysis of the effects of CTNF on cognitive function in healthy adults. Since this was unclear, we have added a description in the introduction. (Lines 60-69).

“For example, previous studies using NIRS reported that the CTNF showed greater beneficial effects on working memory, long-term memory, attention, and executive functions compared to CT alone in healthy adults [18,19]. Small wireless NIRS systems are easy to wear and provide real-time brain activity feedback [20], making them easy to incorporate into cognitive training. However, no systematic review or meta-analysis study has been conducted to investigate effects of CTNF on a wide range of cognitive functions. Summarizing previous findings on which cognitive domains CTNF is effective may support user choice as brain imaging devices become more prevalent and used in the future. Here, we performed a multi-level meta-analysis to reveal the benefit of CTNF on cognitive functions in healthy adults.”

We registered this purpose in PROSPERO. It is difficult to change the registered purpose after completing a meta-analysis. However, we agree with your opination, In the process of selecting literature for our meta-analysis, we found intervention studies combining cognitive training and EEG neurofeedback for children with ADHD. In the study, subjects were children. they were not included in the meta-analysis in this study because we focused on healthy adults. According to your suggestion, we have added a reference to limitation as an area of potential application of CTNF. (Lines 322-325)

“For example, CTNF studies with children with attention deficit/ hypur activity disorder have shown a reduction in their symptoms and problems related to executive function and working memory [50,51], an area in which CTNF is expected to be applied.”

The discussion section needs to be further elaborated and explained at greater length in terms of the research question which the review aims to address, below are some scope that can be further expanded in discussion
-based on the findings, so what? what can be improved? How can this be applied?

Response: We appreciate the reviewer's comment on this point. Following your comments, we have added an advantage of using NIRS for combined training in discussion part. The ease of setup, non-invasiveness, and real-time/instant feedback of the NIRS instrument had the advantage of facilitating combined training like cognitive training. We have added this point. (Lines 303-307)

“It is noted, however, that the NIRS included in this meta-analysis is non-invasive and easy to setup compared with other imaging methods [20,48]. Ease of use and the advantage of not having to increase cognitive training time simply because it is based on real-time or immediate feedback may work well for combined training.”

-based on the studies, what were the common characteristics behind the success or effective intervention-did the intervention period/frequency matter? 

-what were intervention (i.e CTNF) that yield more promising results?

Response: We wish to thank the reviewer for this comment. We agree that this point requires clarification and have added a description focusing on frequency and equipment for imaging as a common feature among the previous studies. (Lines 300-303)

“Two of the studies that had CTNF were short duration but relatively high frequency interventions: 5 or more days per week [18,41]. It is not clear from this study whether the frequency of intervention is optimal in terms of cost to the subject and benefit to cognitive function.”

-Were there differences attributed to patient/age/gender etc?

Response: We thank the reviewer for this comment. The meta-analysis of the cognitive domains (working memory, long-term/episodic memory) revealed positive effect in both older and younger healthy adults. We found little heterogeneity in the results. Due to the small number of the included studies, it is difficult to meta-regression analysis using age. Based on your suggestion, we have added the following discussion. (Lines 314-319)

“Second, we did not perform a subgroup analysis by subject demographics and indicators of cognitive test. The subgroups in which CT is found to be effective are still debated [49]. Although it is important to examine factors such as the effects of aging and participant demographics on the effects of CTNF, meta-regression analysis was not conducted due to the small number of previous studies in our meta-analysis.”

-is findings dependent on NIRS and EEG? or the brain imaging method used?

Response: We wish to thank the reviewer for this comment. In this meta-analysis, all of the included studies used NIRS. On the other hand, we found that one intervention study using EEG also showed improvement in working memory. These findings suggest that CTNF may be effective independent of neurofeedback methods. However, this is the tentative conclusion, and further details will become clearer with the accumulation of future studies. According with your comment, we have added the following discussion. (Lines 312-314)

“Since NIRS was the only instrument used for CTNF in our meta-analysis, it is expected that studies using other imaging instruments would be accumulated.”

Reviewer 2 Report

Dear authors,

Congratulations for your work. You tried to understand the effects of cognitive training  with neuro-feedback on the cognitive function.

1.     Why it is important the aim of your manuscript? Its not clear.

2.     Why you do not performed PICOS criteria?

3.     When you performed the database search? I suggest to perform again and add new databases: Cumulative Index to Nursing and Allied Health (CINAHL), Cochrane Library, EMBASE, PubMed, Scopus, SPORTDiscus, and Web of Science.

4.     Your discussion is too general, try to go more in detail of protocols and debate them.

Author Response

Dear Reviewer

Thank you for reviewing the manuscript.

We have followed your advice and made additions to our introduction, methods and discussion. The following are the revisions. We would appreciate it if you could check them.

Dear authors,

Congratulations for your work. You tried to understand the effects of cognitive training  with neuro-feedback on the cognitive function.

  1. Why it is important the aim of your manuscript? Its not clear.

Response: We wish to thank the reviewer for this comment. In accordance with your suggestion, we have added a description of the significance of the purpose of this study (Lines 60-769).

“For example, previous studies using NIRS reported that the CTNF showed greater beneficial effects on working memory, long-term memory, attention, and executive functions compared to CT alone in healthy adults [18,19]. Small wireless NIRS systems are easy to wear and provide real-time brain activity feedback [20], making them easy to incorporate into cognitive training. However, no systematic review or meta-analysis study has been conducted to investigate effects of CTNF on a wide range of cognitive functions. Summarizing previous findings on which cognitive domains CTNF is effective may support user choice as brain imaging devices become more prevalent and used in the future. Here, we performed a multi-level meta-analysis to reveal the benefit of CTNF on cognitive functions in healthy adults.”

  1. Why you do not performed PICOS criteria?

Response: Thank you for your comments, we have added a description of the PICO critera in 2.2. eligibility criteria. (Lines 79-86)

“Literature eligibility criteria were established based on the PICO criteria [22,23] and literature was searched. Population: healthy adults. Intervention: studies in which both CT and neurofeedback were given to participants. Comparison: Cognitive training or neurofeedback-only interventions. Sham feedback and active control are also included. Outcome: studies assessed cognitive function in participants before and after an intervention period. Research articles reported in academic journals written in English were included; reports on books and conference proceedings were excluded.”

  1. When you performed the database search? I suggest to perform again and add new databases: Cumulative Index to Nursing and Allied Health (CINAHL), Cochrane Library, EMBASE, PubMed, Scopus, SPORTDiscus, and Web of Science.

Response: We thank the reviewer for this comment.Thank you for your comment. The database search date is May 2022, and we added this information in the section 2.3.. (Line 88)

We also ran a search on the search engine you indicated, using the same terms as in the text (CHINAHL: 10 studies; Cochrane library: 102 studies; SPORTDiscus: 6 studies; Web of Science: 106 studies). However, since the relevant references had already been searched in the initial search, we assumed that the initial search was sufficient to cover the article. Therefore, we did not revise the text.

  1. Your discussion is too general, try to go more in detail of protocols and debate them.

Response: We appreciate the reviewer's comment on this point. Following your advice, I made some additions to the discussion section. First, we have added references to frequency and neurofeedback techniques for the intervention included in our meta-analysis. (Lines 300-307)

“Two of the studies that had CTNF were short duration but relatively high frequency interventions: 5 or more days per week [18,41]. It is not clear from this study whether the frequency of intervention is optimal in terms of cost to the subject and benefit to cognitive function. It is noted, however, that the NIRS included in this meta-analysis is non-invasive and easy to setup compared with other imaging methods [20,48]. Ease of use and the advantage of not having to increase cognitive training time simply because it is based on real-time or immediate feedback may work well for combined training.”

In addition, we have made some additional descriptions, especially regarding the limitations of the study. (Lines 308-328)

“This study has some limitations. First, we only included three studies using NIRS for the meta-analysis. Because only three studies were included in the meta-analysis, the reliability and generality of the results is limited. Although heterogeneity was low in the cognitive domains where the CTNF effect was observed, but in general, the small number of data may have biased the results. Since NIRS was the only instrument used for CTNF in our meta-analysis, it is expected that studies using other imaging instruments would be accumulated. Second, we did not perform a subgroup analysis by subject demographics and indicators of cognitive test. The subgroups in which CT is found to be effective are still debated [49]. Although it is important to examine factors such as the effects of aging and participant demographics on the effects of CTNF, meta-regression analysis was not conducted due to the small number of previous studies in our meta-analysis. It is also possible that more detailed results will be obtained when the cognitive test is classified into narrow domain [38] instead of broad domain which was used in this study. Third, our meta-analysis included only healthy adults. It is necessary to examine whether similar results can be obtained for patients or subclinical groups. For example, CTNF studies with children with attention deficit/ hypur activity disorder have shown a reduction in their symptoms and problems related to executive function and working memory [50,51], an area in which CTNF is expected to be applied. Fouth, the results of the quality assessment using the PEDro scale indicated that the studies included in our meta-analysis were of low quality as RCTs. The quality of RCTs examining the effects of CTNF on cognitive function would need to be improved.”

Round 2

Reviewer 2 Report

Dear Authors

you have attended my considerations.

congratulations for the work